# CMOS-Based Implantable Multi-Ion Image Sensor for Mg^2+^ Measurement in the Brain

**DOI:** 10.3390/s25082595

**Published:** 2025-04-20

**Authors:** Yuto Nakamura, Hideo Doi, Yasuyuki Kimura, Tomoko Horio, Yong-Joon Choi, Kazuhiro Takahashi, Toshihiko Noda, Kazuaki Sawada

**Affiliations:** Department of Electrical and Electronic Information Engineering, Toyohashi University of Technology, Toyohashi 441-8122, Aichi, Japan; nakamura.yuto.ry@tut.jp (Y.N.); kimura-y@int.ee.tut.ac.jp (Y.K.); horio.tomoko.av@tut.jp (T.H.); choi.yong.joon.nu@tut.jp (Y.-J.C.); takahashi.kazuhiro.xr@tut.jp (K.T.); noda.toshihiko.zk@tut.jp (T.N.); kazuaki.sawada@tut.jp (K.S.)

**Keywords:** bioimaging, implantable multi-ion image sensor, magnesium ion (Mg^2+^), calcium ion (Ca^2+^), plasticized polyvinyl chloride (PVC) membrane, CMOS array sensor

## Abstract

**Highlights:**

**Abstract:**

An implantable multi-ion image sensor equipped with magnesium ion (Mg^2+^)-and calcium ion (Ca^2+^)-sensitive membranes was fabricated for the selective measurement of extracellular Mg^2+^ in the brain, and the sensor performance was evaluated. This sensor complements the low selectivity of the Mg^2+^-sensitive membrane for Ca^2+^ by depositing a Ca^2+^-sensitive membrane in addition to the Mg^2+^-sensitive membrane on a CMOS (Complementary Metal Oxide Semiconductor)-based potentiometric sensor array with 5.65 × 4.39 µm^2^ pitch, enabling selective measurement of Mg^2+^ and Ca^2+^. Characterization of the sensor confirmed a Ca^2+^ sensitivity of 26.5 mV/dec and Mg^2+^ sensitivity of 19 mV/dec. Based on validation experiments with varying concentrations of Mg^2+^ and Ca^2+^, selective Ca^2+^ and Mg^2+^ measurements were successfully achieved. Furthermore, real-time imaging of Mg^2+^ and Ca^2+^ and quantification of their concentration changes were performed. The developed sensor may be successfully applied for extracellular multi-ion imaging of Mg^2+^ and Ca^2+^ in the living brain.

## 1. Introduction

Neurotransmitters and ions in the brain have been linked to neurological diseases and are essential for understanding brain functions [1,2]. Whereas normal brain functions are performed by the complex interplay of these transmitters, it has been suggested that abnormal transmission and changes in their concentration induce neurological disorders, such as Alzheimer’s disease and epilepsy [3,4].

In recent years, the aging of Japanese society has been one of the most pressing social issues, and the percentage of elderly people in Japan in 2022 was 29.0%. This is well above the 7.0% percentage of elderly people that defines an aging society. Excessive aging is accompanied by an increasing prevalence of memory-related neurological diseases, such as dementia and Parkinson’s disease. Numerous studies have highlighted the association of magnesium ions (Mg^2+^) with dementia, which impairs attention, memory, and psychomotor functions [5,6]. In one of these studies, the intraperitoneal administration of 270 mg/kg magnesium sulfate in rats during inhibitory avoidance training improved their impaired learning ability [7]. However, most of these studies have shown experimental results; the sites and pathways of action of Mg^2+^ are poorly understood, and their mechanisms must be elucidated before they can be applied in biomedicine.

In recent years, imaging studies have been actively conducted to observe the interior of animal brains under in vivo conditions. If extracellular Mg^2+^ in the brains of living individuals can be visualized and analyzed in real time, it is expected to significantly contribute to the elucidation of neurological diseases and drug discovery [8,9]. Fluorescence imaging [10] and nuclear magnetic resonance [11] are mainly used to observe cells in vivo and measure ion concentrations in the brain. However, these techniques have problems such as “difficulty in observing the original activity of living organisms due to labeling”, “difficulty in long-term observation”, “low spatio-temporal resolution”, and “high cost”. Therefore, it is crucial to develop imaging technology to visualize and analyze information in the brains of living individuals in a simple and precise manner.

As an alternative approach, the polyvinyl chloride (PVC) membrane-based Mg^2+^-sensitive electrode [12,13,14] using an ion recognition ionophore has been reported. These electrochemical devices enable the label-free real-time recording of Mg^2+^ response in a solution. Additionally, various types of ions, such as potassium ion (K^+^) [15], calcium ion (Ca^2+^) [16], and sodium ion (Na^+^) [17], can be detected by changing an ionophore corresponding to a target molecule. However, these ion-selective electrodes are not suitable for the spatiotemporal analysis of ion distributions due to a single-point recording.

Our group has developed a 23.55 µm pitch implantable potentiometric biosensor that enables label-free real-time imaging of hydrogen ions (H^+^) distribution [18]. Using this sensor, we previously demonstrated the imaging of changes in extracellular proton dynamics in a living mouse brain [19]. Furthermore, the sensor’s pixel pitch was achieved at 5.65 × 4.39 µm^2^ in recent years, showing a potential for high spatiotemporal H^+^ imaging [20]. Furthermore, we succeeded in measuring calcium ions (Ca^2+^) by immobilizing ionophores on a sensor that selectively detected specific ions [21]. However, the measurement of Mg^2+^, which has been pointed out to be closely related to memory, has not been demonstrated.

In this study, an Mg^2+^-sensitive film that can measure Mg^2+^ and a Ca^2+^-sensitive film that can complement the low selectivity of the Mg^2+^-sensitive film were deposited on an implantable image sensor to fabricate an implantable multi-ion image sensor that can selectively measure Mg^2+^ in the brain. The output characteristics of the fabricated sensor were evaluated. In addition, selective response evaluation experiments were conducted to examine measurement performance.

## 2. Materials and Methods

### 2.1. Materials and Chemicals

Polyvinyl chloride (PVC), 2-Nitrophenyl octyl ether (NPOE), tetrakis-boric acid sodium salt (TFPB), tetrahydrofuran (THF), sodium chloride (99.5%), potassium chloride (99.5%), calcium chloride (95%), magnesium chloride, and D-glucose (98%) were purchased from Wako Pure Chemical Industries Ltd. (Osaka, Japan). Calcium ionophore V (K23E1), sodium, and magnesium ionophore I (ETH 1117) were purchased from Sigma-Aldrich (St. Louis, MO, USA). 4-(2-hydroxyethyl) piperazine-1-ethanesulfonate (HEPES) (>99%) was purchased from Dojindo Laboratories (Tokyo, Japan), and all other reagents were prepared using deionized water (18.2 MΩ) produced with a Milli-Q water system (Tokyo, Japan).

### 2.2. Implantable Ion Image Sensor

Figure 1 shows the general shape and performance of the implantable ion image sensor developed by our research group. This sensor was fabricated by the 0.35 µm CMOS technology. It is designed to be inserted into a living brain and to reduce invasiveness, and the thickness and width of the sensor entry point are 100 µm and 1.1 mm, respectively. As shown in Table 1, it has an array sensor with a pixel size of 5.65 × 4.39 µm ^2^ and a pixel structure of 256 × 32 pixels, which is sufficient for high-resolution imaging, since the neuronal cell body is about 10~30 µm. A Ta_2_O_5_ membrane, functioning as a pH-sensitive film, was deposited onto the array.

In addition, a frame rate of 14.1 fps provides a high spatiotemporal resolution. This study aimed to measure extracellular Mg^2+^ in the living brain by depositing a sensitive membrane containing ionophores that trap specific ions on a potential array sensor.

### 2.3. Proposal of a Multi-Ion Image Sensor

To date, our group has been able to measure ions such as Ca^2+^ and K^+^. Ionophores that can selectively capture the ions to be measured are contained in a polyvinyl chloride (PVC) film solution, and the ion dynamics in the liquid can be measured in real time by depositing the ionophores on the sensor.

Mg^2+^ ionophores that can selectively capture Mg^2+^, the measurement target in this study, also exist. We have been working on the fabrication of an Mg^2+^ image sensor by depositing a Mg^2+^-sensitive film containing these ionophores on the sensing area of an ion image sensor (Figure 2).

We also confirmed that the fabricated Mg^2+^ image sensor is responsive to Mg^2+^. However, the Mg^2+^-sensitive membrane shows high selectivity for brain-coexisting ions such as H^+^, K^+^, and Na^+^ and high response sensitivity to Ca^2+^, which fluctuates in the brain, resulting in inadequate selectivity. This may be because Ca^2+^ is a divalent cation like Mg^2+^, and due to the Hofmeister series, the hydrophilic radius of Mg^2+^ is larger than that of Ca^2+^, making it difficult for the molecular structure of the magnesium ionophore to selectively recognize only Mg^2+^. Therefore, we propose a method in which a Ca^2+^-sensitive membrane that shows high selectivity for coexisting ions in the brain, including Mg^2+^, is painted over the sensing area to form a Mg^2+^-sensitive membrane and a Ca^2+^-sensitive membrane region. Using this method, when the Mg^2+^ concentration changed, only the Mg^2+^-sensitive area responded. When the Ca^2+^ concentration changed, both sensitive areas responded (Figure 3). In other words, the high selectivity of the Ca^2+^-sensitive membrane complements the low selectivity of the Mg^2+^-sensitive membrane for Ca^2+^. It is expected to discriminate between Mg^2+^- and Ca^2+^-dependent potential changes occurring in the Mg^2+^-sensitive membrane.

### 2.4. Sensor Fabrication

In this study, Mg^2+^-and Ca^2+^-sensitized films were formed regionally on a 32 × 256 pixel puncture-type sensor with a pixel pitch of 5.65 × 4.39 µm. Prior to the membrane fabrication, Mg^2+^ and Ca^2+^ membrane solutions were prepared according to previous reports [21,22]. In the Mg^2+^-sensitive membrane solution, 33.9 mg of 2-nitrophenyl octyl ether (NPOE), 3.1 mg of tetrakis-boric acid sodium salt (TFPB), 16.8 mg of PVC, and 3.1 mg of magnesium ionophore (magnesium ionophore I: Sigma-Aldrich) are dissolved in 0.3 mL of tetrahydrofuran (THF). In the Ca^2+^-sensitive membrane solution, 70.6 mg of NPOE, 0.85 mg of TFPB, 28.6 mg of PVC, and 2.85 mg of calcium ionophore (calcium ionophore V: Sigma–Aldrich) were dissolved in 0.4 mL of THF. The amount of THF, which determines the viscosity of the solution, was two to three times lower in the film-sensitive solution used in this study than that used in previous studies. Previous studies showed a dependence between the ion sensitivity and thickness of the sensitized film, and the thickness of the film was made sufficiently thick by repeatedly coating the sensitized film solution. However, when forming two types of sensitive films in a small area, the expansion and mixing of each sensitive area due to overcoating can be problematic; therefore, the viscosity of the sensitive film solution was increased in this study.

The resulting mixtures were applied to an implantable ion image sensor to form sensitive areas, as shown in Figure 4a. After formation, the films were dried at room temperature (25 °C) for 12 h. A micrograph of the sensor chip surface after the PVC membrane formation is shown in Figure 4b. The Mg^2+^- and Ca^2+^-sensitive membranes were separately deposited on the pixel area.

### 2.5. Evaluation of the Ion Response Characteristics in Each Sensitive Region

To confirm the concentration dependence of the Mg^2+^- and Ca^2+^-sensitive regions in the fabricated implantable multi-ion image sensor, measurement solutions were prepared by varying the concentration of each ion in a buffer solution that mimicked the brain. The tip of the fabricated sensor was immersed in a buffer solution for at least 6 h to obtain a sensitive membrane. As shown in Figure 5, the ion response characteristics to changes in the concentrations of sodium ions (Na^+^), K^+^, and H^+^, in addition to Mg^2+^ and Ca^2+^, which are the measurement targets, were measured by changing the sensor every 60 s in a Petri dish filled with a concentration measurement solution. The composition of the base buffer solution is potassium chloride (KCl): 2.5 mM, sodium chloride (NaCl): 150 mM, calcium chloride (CaCl_2_): 2 mM, magnesium chloride (MgCl_2_): 1 mM, glucose: 10 mM, and hydroxyethylpiperazine ethane sulfonic acid (HEPES): 10 mM, and the pH is about 7 pH. The concentration of the ions to be measured varies from 10^−6^ to 10^−1^ M in 1-digit increments, and the pH varies from 7 to 8 pH in 0.5 pH increments.

### 2.6. Simultaneous Measurement of the Mg^2+^ and Ca^2+^ Response

To verify the performance of the implantable multi-ion image sensor in the detection of Mg^2+^ concentration changes and for quantitative measurements, the potential response was measured by varying the concentrations of Ca^2+^ and Mg^2+^ in the buffer solution. The three buffers used were the same as those used in Section 2.3 for the base. Buffer 1 contained MgCl_2_ at 10^−3^ M and CaCl_2_ at 10^−4^ M, buffer 2 contained MgCl_2_ at 10^−2^ M and CaCl_2_ at 10^−4^ M, and buffer 3 contained MgCl_2_ at 10^−2^ M and CaCl_2_ at 10^−3^ M. At the start of the measurement, the sensor was inserted into buffer solution 1. After 100 s, the sensor was transferred to buffer solution 2 and, after another 100 s, to buffer solution 3 (Figure 6). Therefore, it is possible to evaluate the response of this sensor to the Mg^2+^ concentration change in the transition from buffer solution 1 to buffer solution 2 and to the Ca^2+^ concentration change in the transition from buffer solution 2 to buffer solution 3.

## 3. Results and Discussion

### 3.1. Evaluation Results of Ion Response Characteristics in Each Sensing Region

The concentration dependences of the Mg^2+^- and Ca^2+^-sensitive regions for each ion are shown in Figure 7. As shown in Figure 7a, the output voltage (*V*_Out_) for the Mg^2+^-sensitive region increased from 10^−2^ to M and showed slightly lower sensitivity to Mg^2+^ (19 mV/dec). In the concentration range of 10^−4^ to 10^−1^ M Ca^2+^, the sensitivity was measured to be 29.2 mV/dec. This result shows that the Mg^2+^-sensitive membrane has low selectivity for Ca^2+^, as described in Section 2.3. As shown in Figure 7a, the limit of detection for the Mg^2+^-sensitive membrane is higher than 10 mM, because the *V*_Out_ signal of the sensor was saturated less than approximately 10 mM. This indicates that the Mg^2+^ ionophore-entrapped membrane cannot detect extracellular Mg^2+^ fluctuation (~1 mM) due to influence of the interfering ions. Therefore, an improvement of the sensor performance for Mg^2+^ detection is necessary due to the limit of detection for Mg^2+^ being higher than 10 mM at this time. Conversely, *V*_Out_ for the Mg^2+^-sensitive membrane had almost no change for the different concentrations of H^+^, K^+^, and Na^+^. Following these experimental results, further investigation of the Mg^2+^ sensor performance for Ca^2+^ is required for applying physiological conditions. On the other hand, the *V*_Out_ for the Ca^2+^-sensitive region showed a Ca^2+^ response with increasing Ca^2+^ concentrations, with a sensitivity of 26.5 mV/dec, showing high selectivity to H^+^, K^+^, Na^+^, and Mg^2+^ (Figure 7b). The theoretical sensitivity exhibited by the sensitive region when measuring divalent ions is 29.6 mV/dec, according to Nernst’s formula, which tends to decrease the sensitivity. This decrease in sensitivity is thought to be due to ions other than the target ion in the buffer solution (in the case of the Mg^2+^-sensitive region, H^+^, K^+^, Na^+^, and Ca^2+^), which act as interfering ions and prevent the ionophores in the sensitized membrane from capturing ions. This indicates that the Mg^2+^-sensitive region fabricated in this study is susceptible to Ca^2+^. Also, it was ensured that the fabricated sensor functions to measure Mg^2+^ and Ca^2+^ without briefly releasing both membranes from the pixel electrode and can be used basically multiple times in an aqueous solution.

### 3.2. Demonstration Results of Selective Multi-Ion Measurement

To demonstrate the simultaneous imaging of Mg^2+^ and Ca^2+^ responses and the quantitative measurement of the concentration change, visualization experiments were performed using Mg^2+^ and Ca^2+^ solutions. Figure 8a shows the imaging results for the Mg^2+^ and Ca^2+^ responses. The Δ*V*_Out_ of the sensor is shown as an absolute value with a 0.5 V scale by a color bar: the yellow color indicates a high concentration of Mg^2+^ and Ca^2+^, and the blue color indicates a low concentration. The output image of the Mg^2+^-sensitive region changed with the Mg^2+^ concentration. Additionally, the images of both sensitive regions were observed soon after the addition of the Ca^2+^ solution.

Figure 8b shows the biosensor response to changes in the Mg^2+^ and Ca^2+^ concentrations. The *V*_Out_ change (Δ*V*_Out_) in the Mg^2+^-sensitive membrane increased for the concentration change of Mg^2+^, whereas the Δ*V*_Out_ in the Ca^2+^-sensitive membrane hardly changed. After the Ca^2+^ solution was added dropwise, the Δ*V*_Out_ for both the Mg^2+^- and Ca^2+^-sensitive membranes immediately increased, and these sensor signals reached saturated levels.

From these experimental results, and since there was no response less than 10^−2^ M for Mg^2+^, as described in Section 3.1, this originates from the concentration levels of interfering ion Ca^2+^, i.e., it should be considered that the lower detectable Mg^2+^ concentration depends on the concentration of Ca^2+^.

Having confirmed that the discrimination of the measured ion species was successful, we checked whether quantitative concentration derivation was possible from the response to changes in the concentration of each ion. Deriving the Mg^2+^ concentration (*C*_Mg_) and Ca^2+^ concentration (*C*_Ca_) after the concentration change from the potential response (Δ*V*_Mg_ and Δ*V*_Ca_) obtained from the potential response in Figure 8b and the sensor sensitivity (*S*_Mg_ and *S*_Ca_) obtained from the characterization in Section 3.1, the following is obtained.*C*_Mg_ = Δ*V*_Mg_/*S*_Mg_ = 18 mV/19 mV/dec = 8.85 mM*C*_Ca_ = Δ*V*_Ca_/*S*_Ca_ = 26 mV/26.5 mV/dec = 0.98 mM

After the concentration change, the theoretical concentrations were 10 mM for Mg^2+^ and 1 mM for Ca^2+^. The Mg^2+^- and Ca^2+^-sensitive regions were 88.5% and 98%, respectively, in terms of accuracy.

In an experiment to improve memory in mice with a diet containing magnesium, it was reported that the concentration of Mg^2+^ in the brain changed by about 15%. Because the general Mg^2+^ concentration in the brain is 1 mM, it is thought that a minute concentration change of about 0.15 mM is produced [23]. Using this sensor, the concentration change would be about 0.13 mM, assuming an error occurs. This change is detected as a potential change of 1 mV, sufficient to capture the noise of 0.5 mV_rms_ during in vivo measurement.

These results indicated that the Ca^2+^-sensitive region can compensate for the low selectivity for Ca^2+^ in the Mg^2+^-sensitive region. From these experiments and the quantitative evaluation results, the real-time imaging of Mg^2+^ concentration changes, discrimination from Ca^2+^, and quantification of the concentration changes were successfully demonstrated. In addition, a pixel region exposed between the two membrane formation areas functioned as a pH sensor. However, if Mg^2+^ is changing in one part of the sensor and Ca^2+^ in another, the target ion concentration change cannot be measured accurately at this stage, because the two membranes were coated a little too distant in the array. Thus, the development of a membrane patterning technique is required for multi-analyte detection of Mg^2+^ and Ca^2+^ with high spatial resolution toward future bioimaging applications.

Based on our proposed method, the Mg^2+^ and Ca^2+^ imaging and quantitative determination were particularly demonstrated under the specific ion concentration by controlling the Mg^2+^ and Ca^2+^ concentrations. However, we cannot use it to solve the problem, the Mg^2+^-sensitive membrane shows almost no response for real biological Mg^2+^ fluctuations in the presence of Ca^2+^ 2 mM, like the extracellular concentration level. To overcome this problem, improvement in the Mg^2+^ sensor performance is necessary by investigating an ionophore and other materials for biological experiments as future works.

## 4. Conclusions

We fabricated a 32 × 256 pixels implantable multi-ion image sensor using Mg^2+^- and Ca^2+^-sensitive regions. The resulting Mg^2+^-sensitive region on the array showed an output voltage change of 19 mV/dec between 10^−2^ M to 10^−1^ M Mg^2+^, 29.2 mV/dec for Ca^2+^ in the range of 10^−4^ M to 10^−1^ M, and 26.5 mV/dec for Ca^2+^, indicating reasonable sensitivity for the coexisting ions. In the selective response evaluation experiment, the high selectivity of the Ca^2+^-sensitive region complemented the low selectivity of the Mg^2+^-sensitive region, and we verified whether Mg^2+^ concentration changes could be selectively detected. The results showed that the discrimination of Mg^2+^ and Ca^2+^ concentration changes was possible, demonstrating real-time imaging and quantification of Mg^2+^ and Ca^2+^ concentration changes.

Although we succeeded in visualizing changes in Mg^2+^ and Ca^2+^ concentrations on the fabricated sensor device, future investigation for improving the Mg^2+^-sensitive performance is required for biological experiments. If the physiological Mg^2+^ fluctuation of less than 1 mM can be measured, simultaneous Mg^2+^ and Ca^2+^ mapping and multianalyte detection in the brain can be expected by applying inkjet devices and semiconductor technology to paint sensor pixels with a sensitized membrane.

## Figures and Tables

**Figure 1 sensors-25-02595-f001:**
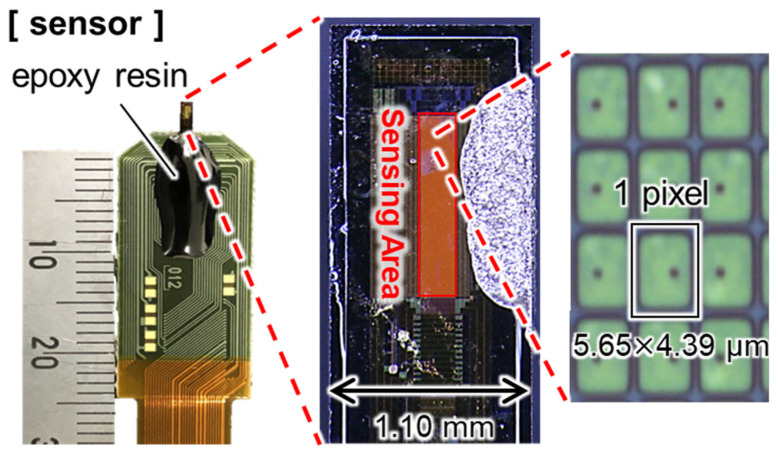
Overview of an implantable pH image sensor.

**Figure 2 sensors-25-02595-f002:**
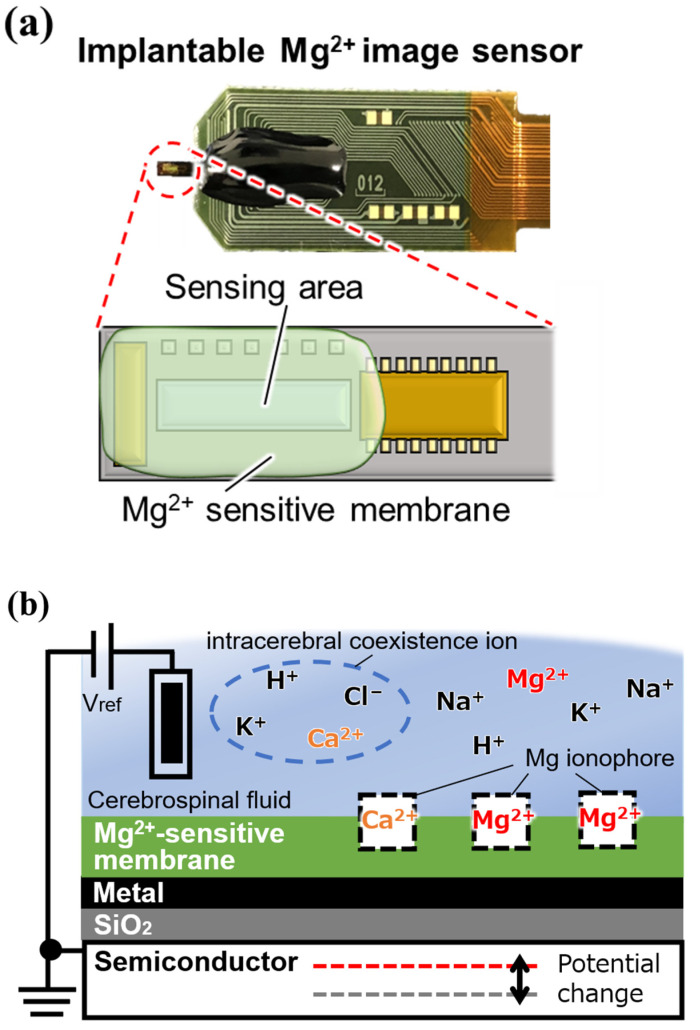
Implantable magnesium ion (Mg^2+^) image sensor: (**a**) overview, and (**b**) cross-sectional view of a single pixel.

**Figure 3 sensors-25-02595-f003:**
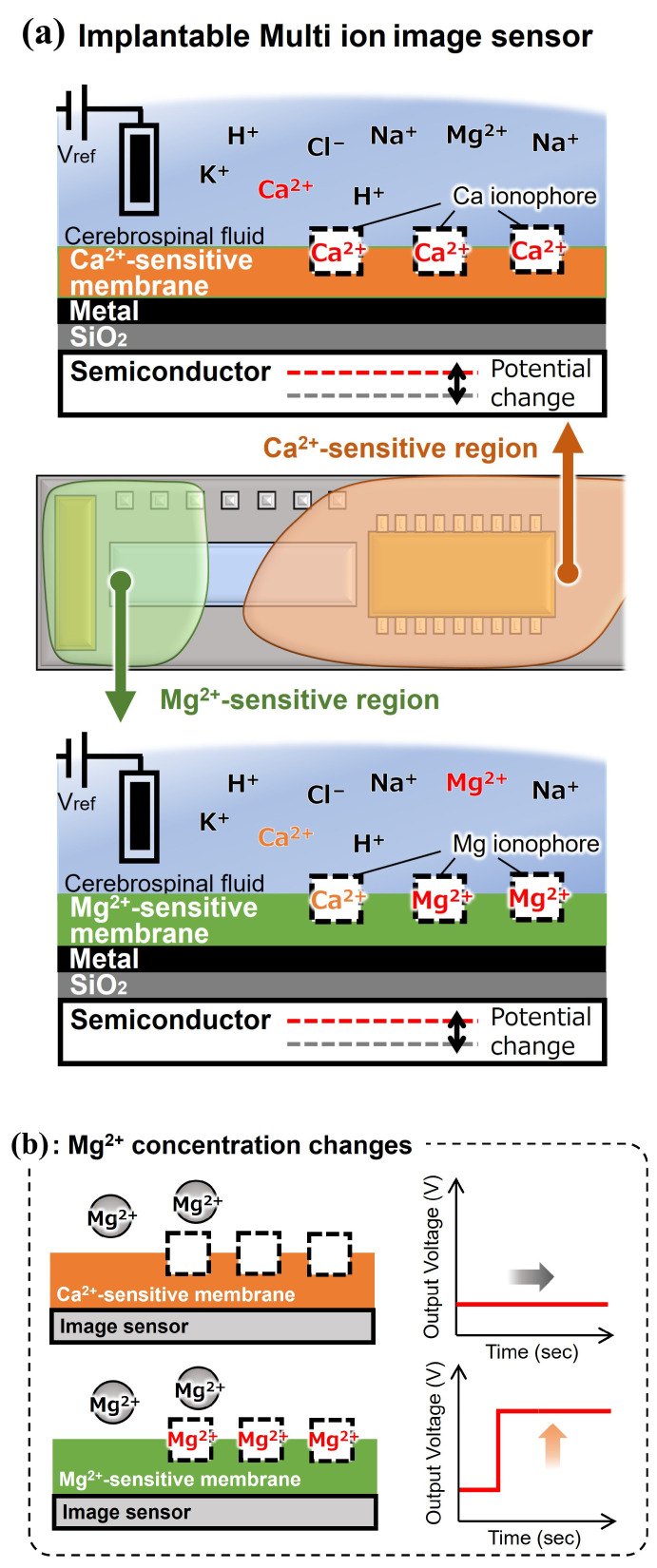
Overview and measurement concept of an implantable multi-ion image sensor: (**a**) outline, (**b**) sensor response to change in the Mg^2+^ concentration, and (**c**) sensor response to change in the calcium ion (Ca^2+^) concentration.

**Figure 4 sensors-25-02595-f004:**
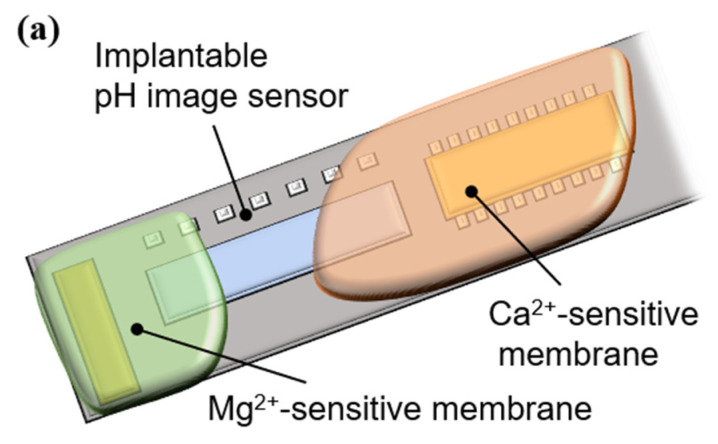
Ion-sensitive membrane deposition: (**a**) outline of an implantable multi-ion image sensor, and (**b**) micrograph of a sensor chip surface.

**Figure 5 sensors-25-02595-f005:**
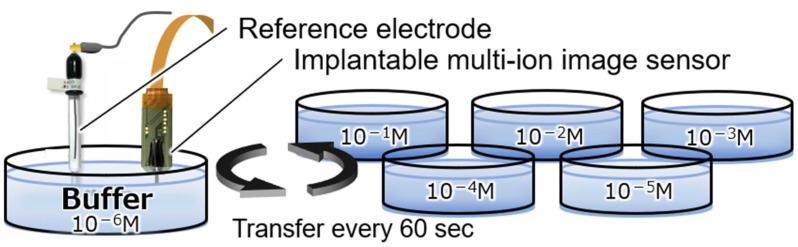
Experimental setup for the measurement of the concentration dependence.

**Figure 6 sensors-25-02595-f006:**
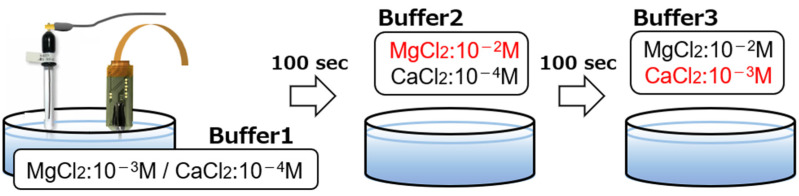
Measurement setup for Mg^2+^ and Ca^2+^ imaging.

**Figure 7 sensors-25-02595-f007:**
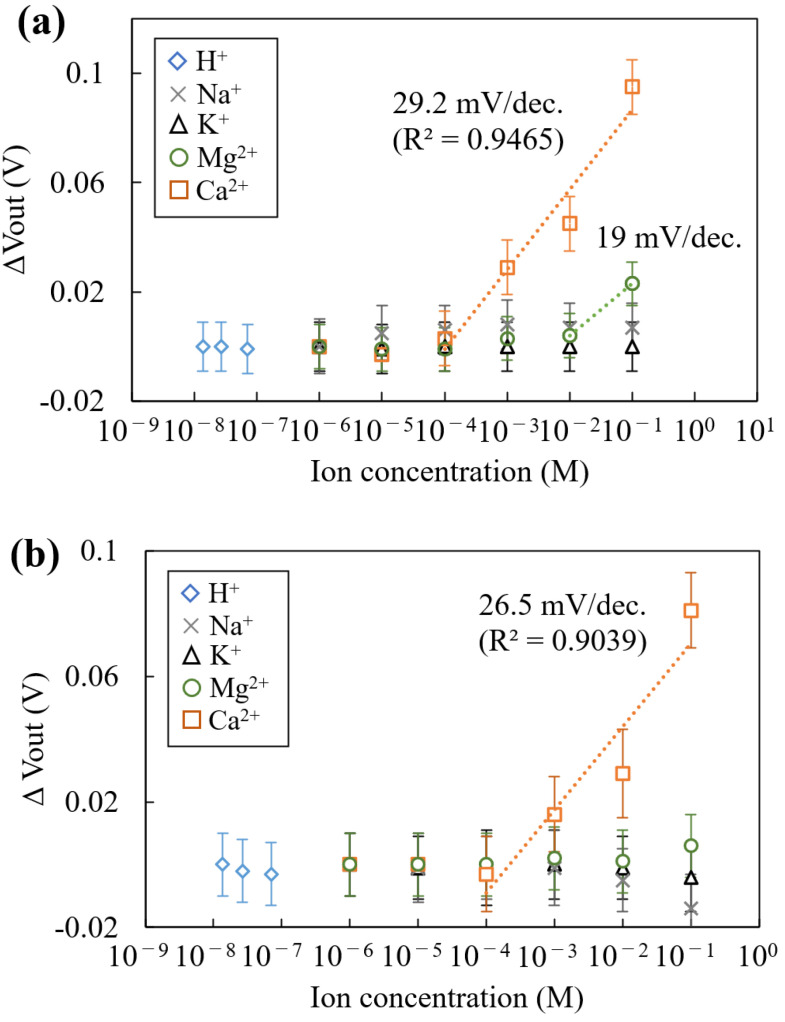
Concentration dependence for the chemicals: (**a**) the sensor response for the Mg^2+^ sensitivity region, and (**b**) the sensor response for the Ca^2+^ sensitivity region. The error bar means the standard deviation of the *V*_Out_ signal obtained from each sensitive region.

**Figure 8 sensors-25-02595-f008:**
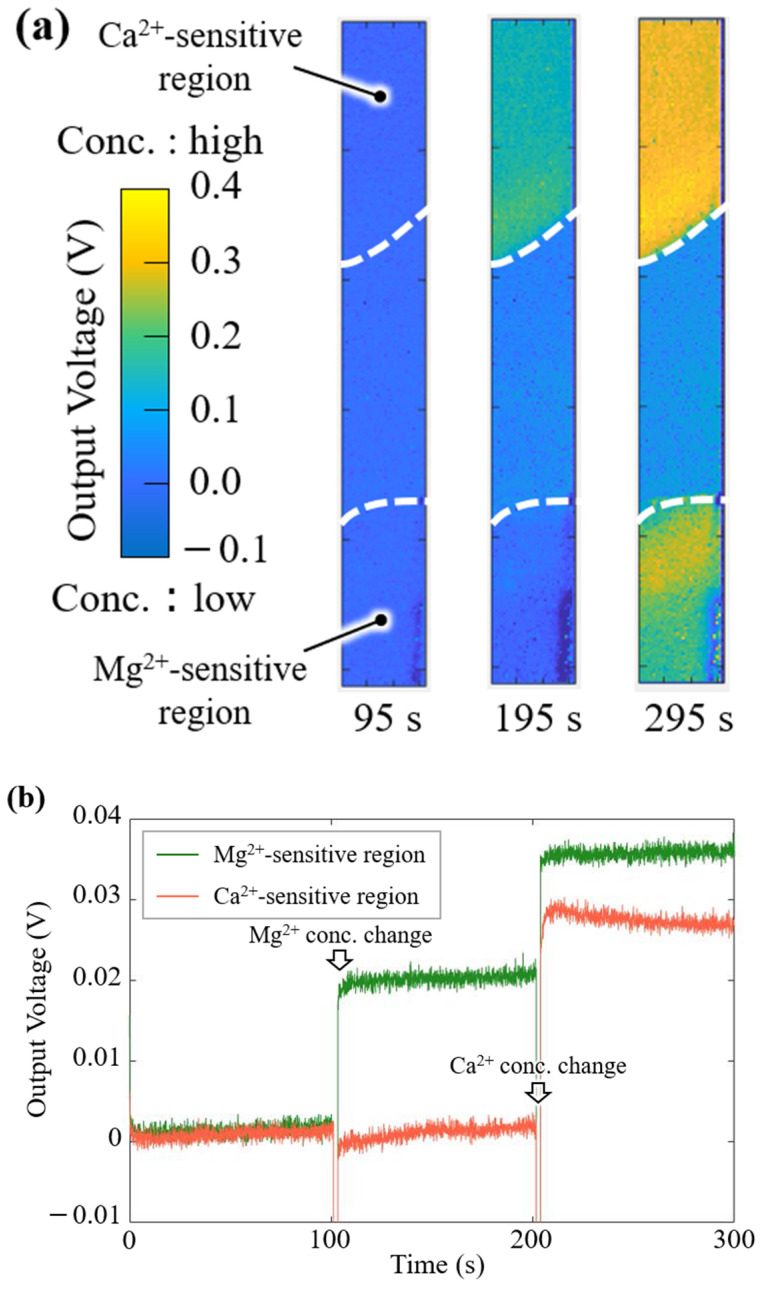
Output response of the implantable multi-ion image sensor: (**a**) two-dimensional images for the Mg^2+^ and Ca^2+^ concentration changes, and (**b**) time course of the sensor output on the Mg^2+^ and Ca^2+^ membranes.

**Table 1 sensors-25-02595-t001:** Details for the sensor specification.

Process	0.35 µm CMOS
Sensor size	9.88 × 1.10 mm^2^
Sensor thickness	100 µm
Sensing area	1124 × 180 µm^2^
Pixel pitch	5.65 × 4.39 µm^2^
Pixel array	256 × 32
Frame rate	14.1 fps

## Data Availability

The data are not publicly available due to privacy or ethical restrictions.

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
