# Peer review of "CMOS-Based Implantable Multi-Ion Image Sensor for Mg2+ Measurement in the Brain"

_sensors, 2025, doi:10.3390/s25082595_

Round 1
Reviewer 1 Report
Comments and Suggestions for Authors
Please see attachment.

The language requires further careful editing.
Author Response
Dear the reviewer 1,
We are sending you our response letter for your review.
Thank you so much for your careful review.
Best regards,
Hideo Doi
Toyohashi University of Technology, Japan

Reviewer 2 Report
Comments and Suggestions for Authors
In this publication, the authors demonstrate an original approach to eliminating the low selectivity of sensors for Mg2+ ions and also demonstrate the possibility of using this sensor for the simultaneous determination of a number of ions.
Despite the positive attitude to the idea, there are a number of comments, without which I do not recommend publishing this material:
1) In electrochemistry, when describing the response of an ion-selective electrode, they traditionally use not the concentrations, but the activities of metal ions in the solution. This characteristic significantly depends on the concentration of electrolytes. Thus, the lack of its consideration leads to a significant distortion of the results in concentrated solutions () it is incorrect to compare the data obtained from a set of buffer solutions with significantly different ionic strengths. I recommend that the authors recalculate the data taking into account the Debye–Hückel theory.
2) The data on the response shown in Figure 7a do not correspond to the experiment shown in Figure 8b. In Figure 7a, when going from [Mg2+]=10^–3 M to 10^–2, the response is <<20 mV, while in Figure 8b, a very large response is observed. Based on the data in Figure 7a, the approach proposed by the authors cannot be used to solve the problem, since there is NO response of 19 mV/dec in the range of 0.001–0.01 M. The authors need to redo the experiment to determine the sensor characteristics. Particular attention should be paid to the sensitivity to magnesium ions - a larger number of experimental points are needed to reliably determine the sensitivity.
3) Do the sensitivity and response time of the sensor depend on the thickness of the ion-selective layer?
4) Can the resulting sensor be used multiple times? How stable are the characteristics in this case? (I recommend demonstrating several cycles of sensor operation)
The work cannot be published in this journal in the present form.
Author Response
Dear the reviewer 2,
We are sending you our response letter for your review.
Thank you so much for your careful review.
Best regards,
Hideo Doi
Toyohashi University of Technology, Japan

Round 2
Reviewer 1 Report
Comments and Suggestions for Authors
Previous comments have been addressed adequately.
Reviewer 2 Report
Comments and Suggestions for Authors
The article can be published in its presented form